# Deep Anomaly Detection with Outlier Exposure

**Dan Hendrycks**
University of California, Berkeley
hendrycks@berkeley.edu

**Mantas Mazeika**
University of Chicago
mantas@ttic.edu

**Thomas Dietterich**
Oregon State University
tgd@oregonstate.edu

## Abstract

It is important to detect anomalous inputs when deploying machine learning systems. The use of larger and more complex inputs in deep learning magnifies the difficulty of distinguishing between anomalous and in-distribution examples. At the same time, diverse image and text data are available in enormous quantities. We propose leveraging these data to improve deep anomaly detection by training anomaly detectors against an auxiliary dataset of outliers, an approach we call Outlier Exposure (OE). This enables anomaly detectors to generalize and detect unseen anomalies. In extensive experiments on natural language processing and small- and large-scale vision tasks, we find that Outlier Exposure significantly improves detection performance. We also observe that cutting-edge generative models trained on CIFAR-10 may assign higher likelihoods to SVHN images than to CIFAR-10 images; we use OE to mitigate this issue. We also analyze the flexibility and robustness of Outlier Exposure, and identify characteristics of the auxiliary dataset that improve performance.

## 1 Introduction

Machine Learning systems in deployment often encounter data that is unlike the model's training data. This can occur in discovering novel astronomical phenomena, finding unknown diseases, or detecting sensor failure. In these situations, models that can detect anomalies (Liu et al., 2018; Emmott et al., 2013) are capable of correctly flagging unusual examples for human intervention, or carefully proceeding with a more conservative fallback policy.

Behind many machine learning systems are deep learning models (Krizhevsky et al., 2012) which can provide high performance in a variety of applications, so long as the data seen at test time is similar to the training data. However, when there is a distribution mismatch, deep neural network classifiers tend to give high confidence predictions on anomalous test examples (Nguyen et al., 2015). This can invalidate the use of prediction probabilities as calibrated confidence estimates (Guo et al., 2017), and makes detecting anomalous examples doubly important.

Several previous works seek to address these problems by giving deep neural network classifiers a means of assigning anomaly scores to inputs. These scores can then be used for detecting *out-of-distribution* (OOD) examples (Hendrycks & Gimpel, 2017; Lee et al., 2018; Liu et al., 2018). These approaches have been demonstrated to work surprisingly well for complex input spaces, such as images, text, and speech. Moreover, they do not require modeling the full data distribution, but instead can use heuristics for detecting unmodeled phenomena. Several of these methods detect unmodeled phenomena by using representations from only in-distribution data.

In this paper, we investigate a complementary method where we train models to detect unmodeled data by learning cues for whether an input is unmodeled. While it is difficult to model the full data distribution, we can learn effective heuristics for detecting out-of-distribution inputs by *exposing* the model to OOD examples, thus learning a more conservative concept of the inliers and enabling the detection of novel forms of anomalies. We propose leveraging diverse, realistic datasets for this purpose, with a method we call Outlier Exposure (OE). OE provides a simple and effective way to consistently improve existing methods for OOD detection.

Through numerous experiments, we extensively evaluate the broad applicability of Outlier Exposure. For multiclass neural networks, we provide thorough results on Computer Vision and Natural

Language Processing tasks which show that Outlier Exposure can help anomaly detectors generalize to and perform well on unseen distributions of outliers, even on large-scale images. We also demonstrate that Outlier Exposure provides gains over several existing approaches to out-of-distribution detection. Our results also show the flexibility of Outlier Exposure, as we can train various models with different sources of outlier distributions. Additionally, we establish that Outlier Exposure can make density estimates of OOD samples significantly more useful for OOD detection. Finally, we demonstrate that Outlier Exposure improves the calibration of neural network classifiers in the realistic setting where a fraction of the data is OOD. Our code is made publicly available at https://github.com/hendrycks/outlier-exposure.

## 2 RELATED WORK

**Out-of-Distribution Detection with Deep Networks.** Hendrycks & Gimpel (2017) demonstrate that a deep, pre-trained classifier has a lower maximum softmax probability on anomalous examples than in-distribution examples, so a classifier can conveniently double as a consistently useful out-of-distribution detector. Building on this work, DeVries & Taylor (2018) attach an auxiliary branch onto a pre-trained classifier and derive a new OOD score from this branch. Liang et al. (2018) present a method which can improve performance of OOD detectors that use a softmax distribution. In particular, they make the maximum softmax probability more discriminative between anomalies and in-distribution examples by pre-processing input data with adversarial perturbations (Goodfellow et al., 2015). Unlike in our work, their parameters are tailored to each source of anomalies.

Lee et al. (2018) train a classifier concurrently with a GAN (Radford et al., 2016; Goodfellow et al., 2014), and the classifier is trained to have lower confidence on GAN samples. For each testing distribution of anomalies, they tune the classifier and GAN using samples from that out-distribution, as discussed in Appendix B of their work. Unlike Liang et al. (2018); Lee et al. (2018), in this work we train our method *without* tuning parameters to fit specific types of anomaly test distributions, so our results are not directly comparable with their results. Many other works (de Vries et al., 2016; Subramanya et al., 2017; Malinin & Gales, 2018; Bevandic et al., 2018) also encourage the model to have lower confidence on anomalous examples. Recently, Liu et al. (2018) provide theoretical guarantees for detecting out-of-distribution examples under the assumption that a suitably powerful anomaly detector is available.

**Utilizing Auxiliary Datasets.** Outlier Exposure uses an auxiliary dataset entirely disjoint from test-time data in order to teach the network better representations for anomaly detection. Goodfellow et al. (2015) train on adversarial examples to increased robustness. Salakhutdinov et al. (2011) pre-train unsupervised deep models on a database of web images for stronger features. Radford et al. (2017) train an unsupervised network on a corpus of Amazon reviews for a month in order to obtain quality sentiment representations. Zeiler & Fergus (2014) find that pre-training a network on the large ImageNet database (Russakovsky et al., 2015) endows the network with general representations that are useful in many fine-tuning applications. Chen & Gupta (2015); Mahajan et al. (2018) show that representations learned from images scraped from the nigh unlimited source of search engines and photo-sharing websites improve object detection performance.

## 3 OUTLIER EXPOSURE

We consider the task of deciding whether or not a sample is from a learned distribution called $\mathcal{D}_{\text{in}}$. Samples from $\mathcal{D}_{\text{in}}$ are called "in-distribution," and otherwise are said to be "out-of-distribution" (OOD) or samples from $\mathcal{D}_{\text{out}}$. In real applications, it may be difficult to know the distribution of outliers one will encounter in advance. Thus, we consider the realistic setting where $\mathcal{D}_{\text{out}}$ is unknown. Given a parametrized OOD detector and an Outlier Exposure (OE) dataset $\mathcal{D}_{\text{out}}^{\text{OE}}$, disjoint from $\mathcal{D}_{\text{out}}^{\text{test}}$, we train the model to discover signals and learn heuristics to detect whether a query is sampled from $\mathcal{D}_{\text{in}}$ or $\mathcal{D}_{\text{out}}^{\text{OE}}$. We find that these heuristics generalize to unseen distributions $\mathcal{D}_{\text{out}}$.

Deep parametrized anomaly detectors typically leverage learned representations from an auxiliary task, such as classification or density estimation. Given a model $f$ and the original learning objective $\mathcal{L}$, we can thus formalize Outlier Exposure as minimizing the objective

$$\mathbb{E}_{(x,y)\sim\mathcal{D}_{\text{in}}}[\mathcal{L}(f(x),y) + \lambda\mathbb{E}_{x'\sim\mathcal{D}_{\text{out}}^{\text{OE}}}[\mathcal{L}_{\text{OE}}(f(x'),f(x),y)]]$$

over the parameters of $f$. In cases where labeled data is not available, then $y$ can be ignored.

Outlier Exposure can be applied with many types of data and original tasks. Hence, the specific formulation of $\mathcal{L}_{\text{OE}}$ is a design choice, and depends on the task at hand and the OOD detector used. For example, when using the maximum softmax probability baseline detector (Hendrycks & Gimpel, 2017), we set $\mathcal{L}_{\text{OE}}$ to the cross-entropy from $f(x')$ to the uniform distribution (Lee et al., 2018). When the original objective $\mathcal{L}$ is density estimation and labels are not available, we set $\mathcal{L}_{\text{OE}}$ to a margin ranking loss on the log probabilities $f(x')$ and $f(x)$.

## 4 EXPERIMENTS

We evaluate OOD detectors with and without OE on a wide range of datasets. Each evaluation consists of an in-distribution dataset $\mathcal{D}_{\text{in}}$ used to train an initial model, a dataset of anomalous examples $\mathcal{D}_{\text{out}}^{\text{OE}}$, and a baseline detector to which we apply OE. We describe the datasets in Section 4.2. The OOD detectors and $\mathcal{L}_{\text{OE}}$ losses are described on a case-by-case basis.

In the first experiment, we show that OE can help detectors generalize to new text and image anomalies. This is all accomplished without assuming access to the test distribution during training or tuning, unlike much previous work. In the confidence branch experiment, we show that OE is flexible and complements a binary anomaly detector. Then we demonstrate that using synthetic outliers does not work as well as using real and diverse data; previously it was assumed that we need synthetic data or carefully selected close-to-distribution data, but real and diverse data is enough. We conclude with experiments in density estimation. In these experiments we find that a cutting-edge density estimator unexpectedly assigns higher density to out-of-distribution samples than in-distribution samples, and we ameliorate this surprising behavior with Outlier Exposure.

### 4.1 EVALUATING OUT-OF-DISTRIBUTION DETECTION METHODS

We evaluate out-of-distribution detection methods on their ability to detect OOD points. For this purpose, we treat the OOD examples as the positive class, and we evaluate three metrics: area under the receiver operating characteristic curve (*AUROC*), area under the precision-recall curve (*AUPR*), and the false positive rate at $N\%$ true positive rate (*FPRN*). The AUROC and AUPR are holistic metrics that summarize the performance of a detection method across multiple thresholds. The AUROC can be thought of as the probability that an anomalous example is given a higher OOD score than a in-distribution example (Davis & Goadrich, 2006). Thus, a higher AUROC is better, and an uninformative detector has an AUROC of 50%. The AUPR is useful when anomalous examples are infrequent (Manning & Schütze, 1999), as it takes the base rate of anomalies into account. During evaluation with these metrics, the base rate of $\mathcal{D}_{\text{out}}^{\text{test}}$ to $\mathcal{D}_{\text{in}}^{\text{test}}$ test examples in all of our experiments is 1:5.

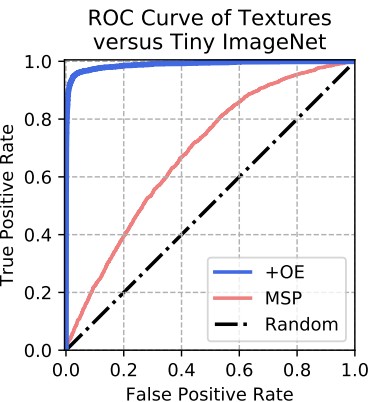

Figure 1: ROC curve with Tiny ImageNet ($\mathcal{D}_{\text{in}}$) and Textures ($\mathcal{D}_{\text{out}}^{\text{test}}$).

Whereas the previous two metrics represent the detection performance across various thresholds, the FPR$N$ metric represents performance at one strict threshold. By observing performance at a strict threshold, we can make clear comparisons among strong detectors. The FPR$N$ metric (Liu et al., 2018; Kumar et al., 2016; Balntas et al., 2016) is the probability that an in-distribution example (negative) raises a false alarm when $N\%$ of anomalous examples (positive) are detected, so a lower FPR$N$ is better. Capturing nearly all anomalies with few false alarms can be of high practical value.

### 4.2 DATASETS

#### 4.2.1 IN-DISTRIBUTION DATASETS

**SVHN.** The SVHN dataset (Netzer et al., 2011) contains $32 \times 32$ color images of house numbers. There are ten classes comprised of the digits 0-9. The training set has $604,388$ images, and the test set has $26,032$ images. For preprocessing, we rescale the pixels to be in the interval $[0, 1]$.
**CIFAR.** The two CIFAR (Krizhevsky & Hinton, 2009) datasets contain $32 \times 32$ natural color images. CIFAR-10 has ten classes while CIFAR-100 has 100. CIFAR-10 and CIFAR-100 classes

are disjoint but have similiarities. For example, CIFAR-10 has "automobiles" and "trucks" but not CIFAR-100's "pickup truck" class. Both have $50,000$ training images and $10,000$ test images. For this and the remaining image datasets, each image is standardized channel-wise.

**Tiny ImageNet.** The Tiny ImageNet dataset (Johnson et al.) is a 200-class subset of the ImageNet (Russakovsky et al., 2015) dataset where images are resized and cropped to $64 \times 64$ resolution. The dataset's images were cropped using bounding box information so that cropped images contain the target, unlike Downsampled ImageNet (Chrabaszcz et al., 2017). The training set has $100,000$ images and the test set has $10,000$ images.

**Places365.** The Places365 training dataset (Zhou et al., 2017) consists in $1,803,460$ large-scale photographs of scenes. Each photograph belongs to one of 365 classes.

**20 Newsgroups.** 20 Newsgroups is a text classification dataset of newsgroup documents with 20 classes and approximately $20,000$ examples split evenly among the classes. We use the standard 60/40 train/test split.

**TREC.** TREC is a question classification dataset with 50 fine-grained classes and $5,952$ individual questions. We reserve 500 examples for the test set, and use the rest for training.

**SST.** The Stanford Sentiment Treebank dataset (Socher et al., 2013) consists of movie reviews expressing positive or negative sentiment. SST has $8,544$ reviews for training and $2,210$ for testing.

### 4.2.2 OUTLIER EXPOSURE DATASETS

**80 Million Tiny Images.** 80 Million Tiny Images (Torralba et al., 2008) is a large-scale, diverse dataset of $32 \times 32$ natural images scrapped from the web. We use this dataset as $\mathcal{D}_{\text{out}}^{\text{OE}}$ for experiments with SVHN, CIFAR-10, and CIFAR-100 as $\mathcal{D}_{\text{in}}$. We remove all examples of 80 Million Tiny Images which appear in the CIFAR datasets, so that $\mathcal{D}_{\text{out}}^{\text{OE}}$ and $\mathcal{D}_{\text{out}}^{\text{test}}$ are disjoint. In Section 5 we note that only a small fraction of this dataset is necessary for successful OE.

**ImageNet-22K.** We use the ImageNet dataset with images from approximately 22 thousand classes as $\mathcal{D}_{\text{out}}^{\text{OE}}$ for Tiny ImageNet and Places365 since images from 80 Million Tiny Images are too low-resolution. To make $\mathcal{D}_{\text{out}}^{\text{OE}}$ and $\mathcal{D}_{\text{out}}^{\text{test}}$ are disjoint, images in ImageNet-1K are removed.

**WikiText-2.** WikiText-2 is a corpus of Wikipedia articles typically used for language modeling. We use WikiText-2 as $\mathcal{D}_{\text{out}}^{\text{OE}}$ for language modeling experiments with Penn Treebank as $\mathcal{D}_{\text{in}}$. For classification tasks on 20 Newsgroups, TREC, and SST, we treat each sentence of WikiText-2 as an individual example, and use simple filters to remove low-quality sentences.

### 4.3 MULTICLASS CLASSIFICATION

In what follows, we use Outlier Exposure to enhance the performance of existing OOD detection techniques with multiclass classification as the original task. Throughout the following experiments, we let $x \in \mathcal{X}$ be a classifier's input and $y \in \mathcal{Y} = \{1, 2, \ldots, k\}$ be a class. We also represent the classifier with the function $f : \mathcal{X} \to \mathbb{R}^k$, such that for any $x$, $\mathbf{1}^\mathsf{T} f(x) = 1$ and $f(x) \succeq 0$.

**Maximum Softmax Probability (MSP).** Consider the maximum softmax probability baseline (Hendrycks & Gimpel, 2017) which gives an input $x$ the OOD score $-\max_c f_c(x)$. Out-of-distribution samples are drawn from various unseen distributions (Appendix A). For each task, we test with approximately twice the number of $\mathcal{D}_{\text{out}}^{\text{test}}$ distributions compared to most other papers, and we also test on NLP tasks. The quality of the OOD example scores are judged with the metrics described in Section 4.1. For this multiclass setting, we perform Outlier Exposure by fine-tuning a pre-trained classifier $f$ so that its posterior is more uniform on $\mathcal{D}_{\text{out}}^{\text{OE}}$ samples. Specifically, the fine-tuning objective is $\mathbb{E}_{(x,y)\sim\mathcal{D}_{\text{in}}}[-\log f_y(x)] + \lambda\mathbb{E}_{x\sim\mathcal{D}_{\text{out}}^{\text{OE}}}[H(\mathcal{U}; f(x))]$, where $H$ is the cross entropy and $\mathcal{U}$ is the uniform distribution over $k$ classes. When there is class imbalance, we could encourage $f(x)$ to match $(P(y = 1), \ldots, P(y = k))$; yet for the datasets we consider, matching $\mathcal{U}$ works well enough. Also, note that training from scratch with OE can result in even better performance than fine-tuning (Appendix C). This approach works on different architectures as well (Appendix D).

Unlike Liang et al. (2018); Lee et al. (2018) and like Hendrycks & Gimpel (2017); DeVries & Taylor (2018), we do not tune our hyperparameters for each $\mathcal{D}_{\text{out}}^{\text{test}}$ distribution, so that $\mathcal{D}_{\text{out}}^{\text{test}}$ is kept unknown like with real-world anomalies. Instead, the $\lambda$ coefficients were determined early in experimentation with validation $\mathcal{D}_{\text{out}}^{\text{val}}$ distributions described in Appendix A. In particular, we use $\lambda = 0.5$ for vision experiments and $\lambda = 1.0$ for NLP experiments. Like previous OOD detection methods involving network fine-tuning, we chose $\lambda$ so that impact on classification accuracy is negligible.

For nearly all of the vision experiments, we train Wide Residual Networks (Zagoruyko & Komodakis, 2016) and then fine-tune network copies with OE for 10 epochs. However we use a pretrained ResNet-18 for Places365. For NLP experiments, we train 2-layer GRUs (Cho et al., 2014) for 5 epochs, then fine-tune network copies with OE for 2 epochs. Networks trained on CIFAR-10 or CIFAR-100 are exposed to images from 80 Million Tiny Images, and the Tiny ImageNet and Places365 classifiers are exposed to ImageNet-22K. NLP classifiers are exposed to WikiText-2. Further architectural and training details are in Appendix B. For all tasks, OE improves average performance by a large margin. Averaged results are shown in Tables 1 and 2. Sample ROC curves are shown in Figures 1 and 4. Detailed results on individual $\mathcal{D}_{\text{out}}^{\text{test}}$ datasets are in Table 7 and Table 8 in Appendix A. Notice that the SVHN classifier with OE can be used to detect new anomalies such as emojis and street view alphabet letters, even though $\mathcal{D}_{\text{OE}}^{\text{test}}$ is a dataset of natural images. Thus, Outlier Exposure helps models to generalize to unseen $\mathcal{D}_{\text{out}}^{\text{test}}$ distributions far better than the baseline.

| | FPR95 ↓ | | AUROC ↑ | | AUPR ↑ | |
| $\mathcal{D}_{\text{in}}$ | MSP | +OE | MSP | +OE | MSP | +OE |
|---|---|---|---|---|---|---|
| SVHN | 6.3 | 0.1 | 98.0 | 100.0 | 91.1 | 99.9 |
| CIFAR-10 | 34.9 | 9.5 | 89.3 | 97.8 | 59.2 | 90.5 |
| CIFAR-100 | 62.7 | 38.5 | 73.1 | 87.9 | 30.1 | 58.2 |
| Tiny ImageNet | 66.3 | 14.0 | 64.9 | 92.2 | 27.2 | 79.3 |
| Places365 | 63.5 | 28.2 | 66.5 | 90.6 | 33.1 | 71.0 |

Table 1: Out-of-distribution image detection for the maximum softmax probability (MSP) baseline detector and the MSP detector after fine-tuning with Outlier Exposure (OE). Results are percentages and also an average of 10 runs. Expanded results are in Appendix A.

| | FPR90 ↓ | | AUROC ↑ | | AUPR ↑ | |
| $\mathcal{D}_{\text{in}}$ | MSP | +OE | MSP | +OE | MSP | +OE |
|---|---|---|---|---|---|---|
| 20 Newsgroups | 42.4 | 4.9 | 82.7 | 97.7 | 49.9 | 91.9 |
| TREC | 43.5 | 0.8 | 82.1 | 99.3 | 52.2 | 97.6 |
| SST | 74.9 | 27.3 | 61.6 | 89.3 | 22.9 | 59.4 |

Table 2: Comparisons between the MSP baseline and the MSP of the natural language classifier fine-tuned with OE. Results are percentages and averaged over 10 runs.

**Confidence Branch.**    A recently proposed OOD detection technique (DeVries & Taylor, 2018) involves appending an OOD scoring branch $b : \mathcal{X} \rightarrow [0, 1]$ onto a deep network. Trained with samples from only $\mathcal{D}_{\text{in}}$, this branch estimates the network's confidence on any input. The creators of this technique made their code publicly available, so we use their code to train new 40-4 Wide Residual Network classifiers. We fine-tune the confidence branch with Outlier Exposure by adding $0.5\mathbb{E}_{x \sim \mathcal{D}_{\text{out}}^{\text{OE}}}[\log b(x)]$ to the network's original optimization objective. In Table 3, the baseline values are derived from the maximum softmax probabilities produced by the classifier trained with DeVries & Taylor (2018)'s publicly available training code. The confidence branch improves over this MSP detector, and after OE, the confidence branch detects anomalies more effectively.

| | FPR95 ↓ | | | AUROC ↑ | | | AUPR ↑ | | |
| $\mathcal{D}_{\text{in}}$ | MSP | Branch | +OE | MSP | Branch | +OE | MSP | Branch | +OE |
|---|---|---|---|---|---|---|---|---|---|
| CIFAR-10 | 49.3 | 38.7 | 20.8 | 84.4 | 86.9 | 93.7 | 51.9 | 48.6 | 66.6 |
| CIFAR-100 | 55.6 | 47.9 | 42.0 | 77.6 | 81.2 | 85.5 | 36.5 | 44.4 | 54.7 |
| Tiny ImageNet | 64.3 | 66.9 | 20.1 | 65.3 | 63.4 | 90.6 | 30.3 | 25.7 | 75.2 |

Table 3: Comparison among the maximum softmax probability, Confidence Branch, and Confidence Branch + OE OOD detectors. The same network architecture is used for all three detectors. All results are percentages, and averaged across all $\mathcal{D}_{\text{out}}^{\text{test}}$ datasets.

**Synthetic Outliers.**    Outlier Exposure leverages the simplicity of downloading real datasets, but it is possible to generate synthetic outliers. Note that we made an attempt to distort images with noise and use these as outliers for OE, but the classifier quickly memorized this statistical pattern and did not detect new OOD examples any better than before (Hafner et al., 2018). A method with better

success is from Lee et al. (2018). They carefully train a GAN to generate synthetic examples near the classifier's decision boundary. The classifier is encouraged to have a low maximum softmax probability on these synthetic examples. For CIFAR classifiers, they mention that a GAN can be a better source of anomalies than datasets such as SVHN. In contrast, we find that the simpler approach of drawing anomalies from a diverse dataset is sufficient for marked improvements in OOD detection.

We train a 40-4 Wide Residual Network using Lee et al. (2018)'s publicly available code, and use the network's maximum softmax probabilities as our baseline. Another classifier trains concurrently with a GAN so that the classifier assigns GAN-generated examples a high OOD score. We want each $\mathcal{D}_{\text{out}}^{\text{test}}$ to be novel. Consequently we use their code's default hyperparameters, and exactly one model encounters all tested $\mathcal{D}_{\text{out}}^{\text{test}}$ distributions. This is unlike their work since, for each $\mathcal{D}_{\text{out}}^{\text{test}}$ distribution, they train and tune a new network. We do not evaluate on Tiny ImageNet, Places365, nor text, since DCGANs cannot stably generate such images and text reliably. Lastly, we take the network trained in tandem with a GAN and fine-tune it with OE. Table 4 shows the large gains from using OE with a real and diverse dataset over using synthetic samples from a GAN.

| $\mathcal{D}_{\text{in}}$ | FPR95 ↓ | | | AUROC ↑ | | | AUPR ↑ | | |
|---|---|---|---|---|---|---|---|---|---|
| | MSP | +GAN | +OE | MSP | +GAN | +OE | MSP | +GAN | +OE |
| CIFAR-10 | 32.3 | 37.3 | 11.8 | 88.1 | 89.6 | 97.2 | 51.1 | 59.0 | 88.5 |
| CIFAR-100 | 66.6 | 66.2 | 49.0 | 67.2 | 69.3 | 77.9 | 27.4 | 33.0 | 44.7 |

Table 4: Comparison among the maximum softmax probability (MSP), MSP + GAN, and MSP + GAN + OE OOD detectors. The same network architecture is used for all three detectors. All results are percentages and averaged across all $\mathcal{D}_{\text{out}}^{\text{test}}$ datasets.

## 4.4 DENSITY ESTIMATION

Density estimators learn a probability density function over the data distribution $\mathcal{D}_{\text{in}}$. Anomalous examples should have low probability density, as they are scarce in $\mathcal{D}_{\text{in}}$ by definition (Nalisnick et al., 2019). Consequently, density estimates are another means by which to score anomalies (Zong et al., 2018). We show the ability of OE to improve density estimates on low-probability, outlying data.

**PixelCNN++.** Autoregressive neural density estimators provide a way to parametrize the probability density of image data. Although sampling from these architectures is slow, they allow for evaluating the probability density with a single forward pass through a CNN, making them promising candidates for OOD detection. We use PixelCNN++ (Salimans et al., 2017) as a baseline OOD detector, and we train it on CIFAR-10. The OOD score of example $x$ is the bits per pixel (BPP), defined as $\text{nll}(x)/\texttt{num\_pixels}$, where nll is the negative log-likelihood. With this loss we fine-tune for 2 epochs using OE, which we find is sufficient for the training loss to converge. Here OE is implemented with a margin loss over the log-likelihood difference between in-distribution and anomalous examples, so that the loss for a sample $x_{\text{in}}$ from $\mathcal{D}_{\text{in}}$ and point $x_{\text{out}}$ from $\mathcal{D}_{\text{out}}^{\text{OE}}$ is

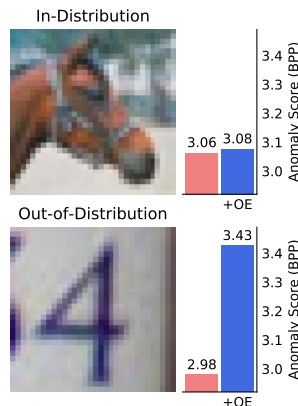

Figure 2: OOD scores from PixelCNN++ on images from CIFAR-10 and SVHN.

$$\max\{0, \texttt{num\_pixels} + \text{nll}(x_{\text{in}}) - \text{nll}(x_{\text{out}})\}.$$

Results are shown in Table 5. Notice that PixelCNN++ without OE unexpectedly assigns lower BPP from SVHN images than CIFAR-10 images. For all $\mathcal{D}_{\text{out}}^{\text{test}}$ datasets, OE significantly improves results.

**Language Modeling.** We next explore using OE on language models. We use QRNN (Merity et al., 2018a;b) language models as baseline OOD detectors. For the OOD score, we use bits per character (BPC) or bits per word (BPW), defined as $\text{nll}(x)/\texttt{sequence\_length}$, where $\text{nll}(x)$ is the negative log-likelihood of the sequence $x$. Outlier Exposure is implemented by adding the cross entropy to the uniform distribution on tokens from sequences in $\mathcal{D}_{\text{out}}^{\text{OE}}$ as an additional loss term.

For $\mathcal{D}_{\text{in}}$, we convert the language-modeling version of Penn Treebank, split into sequences of length 70 for backpropagation for word-level models, and 150 for character-level models. We do not train or evaluate with preserved hidden states as in BPTT. This is because retaining hidden states would

| $\mathcal{D}_{in}$ | $\mathcal{D}_{out}^{test}$ | FPR95 ↓ | | AUROC ↑ | | AUPR ↑ | |
|---|---|---|---|---|---|---|---|
| | | BPP | +OE | BPP | +OE | BPP | +OE |
| | Gaussian | 0.0 | 0.0 | 100.0 | 100.0 | 100.0 | 99.6 |
| | Rademacher | 61.4 | 50.3 | 44.2 | 56.5 | 14.2 | 17.3 |
| | Blobs | 17.2 | 1.3 | 93.2 | 99.5 | 60.0 | 96.2 |
| CIFAR-10 | Textures | 96.8 | 48.9 | 69.4 | 88.8 | 40.9 | 70.0 |
| | SVHN | 98.8 | 86.9 | 15.8 | 75.8 | 9.7 | 60.0 |
| | Places365 | 86.1 | 50.3 | 74.8 | 89.3 | 38.6 | 70.4 |
| | LSUN | 76.9 | 43.2 | 76.4 | 90.9 | 36.5 | 72.4 |
| | CIFAR-100 | 96.1 | 89.8 | 52.4 | 68.5 | 19.0 | 41.9 |
| Mean | | 66.6 | **46.4** | 65.8 | **83.7** | 39.9 | **66.0** |

Table 5: OOD detection results with a PixelCNN++ density estimator, and the same estimator after applying OE. The model's bits per pixel (BPP) scores each sample. All results are percentages. Test distributions $\mathcal{D}_{out}^{test}$ are described in Appendix A.

greatly simplify the task of OOD detection. Accordingly, the OOD detection task is to provide a score for 70- or 150-token sequences in the unseen $\mathcal{D}_{out}^{test}$ datasets.

We train word-level models for 300 epochs, and character-level models for 50 epochs. We then fine-tune using OE on WikiText-2 for 5 epochs. For the character-level language model, we create a character-level version of WikiText-2 by converting words to lowercase and leaving out characters which do not appear in PTB. OOD detection results for the word-level and character-level language models are shown in Table 6; expanded results and $\mathcal{D}_{out}^{test}$ descriptions are in Appendix F. In all cases, OE improves over the baseline, and the improvement is especially large for the word-level model.

| $\mathcal{D}_{in}$ | FPR90 ↓ | | AUROC ↑ | | AUPR ↑ | |
|---|---|---|---|---|---|---|
| | BPC/BPW | +OE | BPC/BPW | +OE | BPC/BPW | +OE |
| PTB Characters | 99.0 | 89.4 | 77.5 | 86.3 | 76.0 | 86.7 |
| PTB Words | 48.5 | 0.98 | 81.2 | 99.2 | 44.0 | 97.8 |

Table 6: OOD detection results on Penn Treebank language models. Results are percentages averaged over the $\mathcal{D}_{out}^{test}$ datasets. Expanded results are in Appendix F.

## 5 DISCUSSION

**Extensions to Multilabel Classifiers and the Reject Option.** Outlier Exposure can work in more classification regimes than just those considered above. For example, a multi*label* classifier trained on CIFAR-10 obtains an 88.8% mean AUROC when using the maximum prediction probability as the OOD score. By training with OE to decrease the classifier's output probabilities on OOD samples, the mean AUROC increases to 97.1%. This is slightly less than the AUROC for a multiclass model tuned with OE. An alternative OOD detection formulation is to give classifiers a "reject class" (Bartlett & Wegkamp, 2008). Outlier Exposure is also flexible enough to improve performance in this setting, but we find that even with OE, classifiers with the reject option or multilabel outputs are not as competitive as OOD detectors with multiclass outputs.

**Flexibility in Choosing $\mathcal{D}_{out}^{OE}$.** Early in experimentation, we found that the choice of $\mathcal{D}_{out}^{OE}$ is important for generalization to unseen $\mathcal{D}_{out}^{test}$ distributions. For example, adding Gaussian noise to samples from $\mathcal{D}_{in}$ to create $\mathcal{D}_{out}^{OE}$ does not teach the network to generalize to unseen anomaly distributions for complex $\mathcal{D}_{in}$. Similarly, we found in Section 4.3 that synthetic anomalies do not work as well as real data for $\mathcal{D}_{out}^{OE}$. In contrast, our experiments demonstrate that the large datasets of realistic anomalies described in Section 4.2.2 do generalize to unseen $\mathcal{D}_{out}^{test}$ distributions.

In addition to size and realism, we found diversity of $\mathcal{D}_{out}^{OE}$ to be an important factor. Concretely, a CIFAR-100 classifier with CIFAR-10 as $\mathcal{D}_{out}^{OE}$ hardly improves over the baseline. A CIFAR-10 classifier exposed to ten CIFAR-100 outlier classes corresponds to an average AUPR of 78.5%. Exposed to 30 such classes, the classifier's average AUPR becomes 85.1%. Next, 50 classes corresponds to 85.3%, and from thereon additional CIFAR-100 classes barely improve performance. This suggests that dataset diversity is important, not just size. In fact, experiments in this paper often used around

1% of the images in the 80 Million Tiny Images dataset since we only briefly fine-tuned the models. We also found that using only 50,000 examples from this dataset led to a negligible degradation in detection performance. Additionally, $\mathcal{D}_{out}^{OE}$ datasets with significantly different statistics can perform similarly. For instance, using the Project Gutenberg dataset in lieu of WikiText-2 for $\mathcal{D}_{out}^{OE}$ in the SST experiments gives an average AUROC of 90.1% instead of 89.3%.

**Closeness of $\mathcal{D}_{out}^{test}$, $\mathcal{D}_{out}^{OE}$, and $\mathcal{D}_{in}^{test}$.** Our experiments show several interesting effects of the closeness of the datasets involved. Firstly, we find that $\mathcal{D}_{out}^{test}$ and $\mathcal{D}_{out}^{OE}$ need not be close for training with OE to improve performance on $\mathcal{D}_{out}^{test}$. In Appendix A, we observe that an OOD detector for SVHN has its performance improve with Outlier Exposure even though (1) $\mathcal{D}_{out}^{OE}$ samples are images of natural scenes rather than digits, and (2) $\mathcal{D}_{out}^{test}$ includes unnatural examples such as emojis. We observed the same in our preliminary experiments with MNIST; using 80 Million Tiny Images as $\mathcal{D}_{out}^{OE}$, OE increased the AUPR from 94.2% to 97.0%.

Secondly, we find that the closeness of $\mathcal{D}_{out}^{OE}$ to $\mathcal{D}_{in}^{test}$ can be an important factor in the success of OE. In the NLP experiments, preprocessing $\mathcal{D}_{out}^{OE}$ to be closer to $\mathcal{D}_{in}$ improves OOD detection performance significantly. Without preprocessing, the network may discover easy-to-learn cues which reveal whether the input is in- or out-of-distribution, so the OE training objective can be optimized in unintended ways. That results in weaker detectors. In a separate experiment, we use Online Hard Example Mining so that difficult outliers have more weight in Outlier Exposure. Although this improves performance on the hardest anomalies, anomalies without plausible local statistics like noise are detected slightly less effectively than before. Thus hard or close-to-distribution examples do not necessarily teach the detector all valuable heuristics for detecting various forms of anomalies. Real-world applications of OE could use the method of Sun et al. (2018) to refine a scraped $\mathcal{D}_{out}^{OE}$ auxiliary dataset to be appropriately close to $\mathcal{D}_{in}^{test}$.

**OE Improves Calibration.** When using classifiers for prediction, it is important that confidence estimates given for the predictions do not misrepresent empirical performance. A calibrated classifier gives confidence probabilities that match the empirical frequency of correctness. That is, if a calibrated model predicts an event with 30% probability, then 30% of the time the event transpires.

Existing confidence calibration approaches consider the standard setting where data at test-time is always drawn from $\mathcal{D}_{in}$. We extend this setting to include examples from $\mathcal{D}_{out}^{test}$ at test-time since systems should provide calibrated probabilities on both in- and out-of-distribution samples. The classifier should have low-confidence predictions on these OOD examples, since they do not have a class. Building on the temperature tuning method of Guo et al. (2017), we demonstrate that

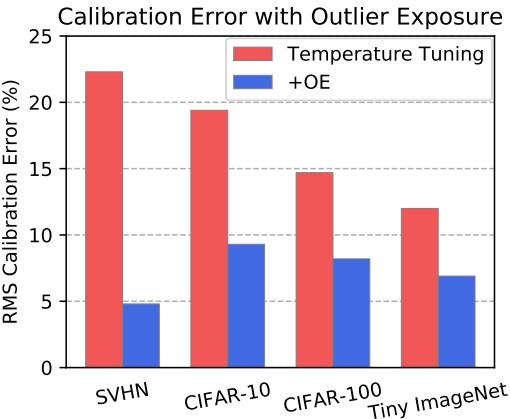

Figure 3: Root Mean Square Calibration Error values with temperature tuning and temperature tuning + OE across various datasets.

OE can improve calibration performance in this realistic setting. Summary results are shown in Figure 3. Detailed results and a description of the metrics are in Appendix G.

## 6 CONCLUSION

In this paper, we proposed Outlier Exposure, a simple technique that enhances many current OOD detectors across various settings. It uses out-of-distribution samples to teach a network heuristics to detect new, unmodeled, out-of-distribution examples. We showed that this method is broadly applicable in vision and natural language settings, even for large-scale image tasks. OE can improve model calibration and several previous anomaly detection techniques. Further, OE can teach density estimation models to assign more plausible densities to out-of-distribution samples. Finally, Outlier Exposure is computationally inexpensive, and it can be applied with low overhead to existing systems. In summary, Outlier Exposure is an effective and complementary approach for enhancing out-of-distribution detection systems.

ACKNOWLEDGMENTS

We thank NVIDIA for donating GPUs used in this research. This research was supported by a grant from the Future of Life Institute.

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

# A    EXPANDED MULTICLASS RESULTS

Expanded mutliclass out-of-distribution detection results are in Table 7 and Table 8.

| $\mathcal{D}_{in}$ | $\mathcal{D}_{out}^{test}$ | FPR95 ↓ | | AUROC ↑ | | AUPR ↑ | |
|---|---|---|---|---|---|---|---|
| | | MSP | +OE | MSP | +OE | MSP | +OE |
| SVHN | Gaussian | 5.4 | 0.0 | 98.2 | 100. | 90.5 | 100. |
| | Bernoulli | 4.4 | 0.0 | 98.6 | 100. | 91.9 | 100. |
| | Blobs | 3.7 | 0.0 | 98.9 | 100. | 93.5 | 100. |
| | Icons-50 | 11.4 | 0.3 | 96.4 | 99.8 | 87.2 | 99.2 |
| | Textures | 7.2 | 0.2 | 97.5 | 100. | 90.9 | 99.7 |
| | Places365 | 5.6 | 0.1 | 98.1 | 100. | 92.5 | 99.9 |
| | LSUN | 6.4 | 0.1 | 97.8 | 100. | 91.0 | 99.9 |
| | CIFAR-10 | 6.0 | 0.1 | 98.0 | 100. | 91.2 | 99.9 |
| | Chars74K | 6.4 | 0.1 | 97.9 | 100. | 91.5 | 99.9 |
| | Mean | 6.28 | **0.07** | 97.95 | **99.96** | 91.12 | **99.85** |
| CIFAR-10 | Gaussian | 14.4 | 0.7 | 94.7 | 99.6 | 70.0 | 94.3 |
| | Rademacher | 47.6 | 0.5 | 79.9 | 99.8 | 32.3 | 97.4 |
| | Blobs | 16.2 | 0.6 | 94.5 | 99.8 | 73.7 | 98.9 |
| | Textures | 42.8 | 12.2 | 88.4 | 97.7 | 58.4 | 91.0 |
| | SVHN | 28.8 | 4.8 | 91.8 | 98.4 | 66.9 | 89.4 |
| | Places365 | 47.5 | 17.3 | 87.8 | 96.2 | 57.5 | 87.3 |
| | LSUN | 38.7 | 12.1 | 89.1 | 97.6 | 58.6 | 89.4 |
| | CIFAR-100 | 43.5 | 28.0 | 87.9 | 93.3 | 55.8 | 76.2 |
| | Mean | 34.94 | **9.50** | 89.27 | **97.81** | 59.16 | **90.48** |
| CIFAR-100 | Gaussian | 54.3 | 12.1 | 64.7 | 95.7 | 19.7 | 71.1 |
| | Rademacher | 39.0 | 17.1 | 79.4 | 93.0 | 30.1 | 56.9 |
| | Blobs | 58.0 | 12.1 | 75.3 | 97.2 | 29.7 | 86.2 |
| | Textures | 71.5 | 54.4 | 73.8 | 84.8 | 33.3 | 56.3 |
| | SVHN | 69.3 | 42.9 | 71.4 | 86.9 | 30.7 | 52.9 |
| | Places365 | 70.4 | 49.8 | 74.2 | 86.5 | 33.8 | 57.9 |
| | LSUN | 74.0 | 57.5 | 70.7 | 83.4 | 28.8 | 51.4 |
| | CIFAR-10 | 64.9 | 62.1 | 75.4 | 75.7 | 34.3 | 32.6 |
| | Mean | 62.66 | **38.50** | 73.11 | **87.89** | 30.05 | **58.15** |
| Tiny ImageNet | Gaussian | 72.6 | 45.4 | 33.7 | 76.5 | 12.3 | 28.6 |
| | Rademacher | 51.7 | 49.0 | 62.0 | 65.1 | 18.8 | 20.0 |
| | Blobs | 79.4 | 0.0 | 48.2 | 100. | 14.4 | 99.9 |
| | Textures | 76.4 | 4.8 | 70.4 | 98.5 | 31.4 | 95.8 |
| | SVHN | 52.3 | 0.4 | 80.8 | 99.8 | 42.8 | 98.2 |
| | Places365 | 63.6 | 0.4 | 76.9 | 99.8 | 36.3 | 99.3 |
| | LSUN | 67.0 | 0.4 | 74.2 | 99.9 | 31.2 | 99.5 |
| | ImageNet | 67.3 | 11.6 | 72.8 | 97.9 | 30.0 | 92.9 |
| | Mean | 66.27 | **13.99** | 64.86 | **92.18** | 27.15 | **79.26** |
| Places365 | Gaussian | 37.1 | 9.4 | 72.2 | 93.5 | 23.5 | 54.1 |
| | Rademacher | 60.4 | 13.5 | 47.7 | 90.2 | 14.6 | 44.9 |
| | Blobs | 73.7 | 0.1 | 41.9 | 100.0 | 13.0 | 99.4 |
| | Icons-50 | 59.1 | 0.0 | 82.7 | 99.9 | 49.9 | 99.7 |
| | Textures | 84.1 | 49.9 | 66.6 | 91.4 | 24.6 | 75.7 |
| | SVHN | 19.9 | 0.0 | 96.6 | 100.0 | 90.5 | 99.9 |
| | ImageNet | 86.3 | 65.3 | 63.0 | 86.5 | 25.1 | 69.7 |
| | Places69 | 87.3 | 87.5 | 61.5 | 63.1 | 23.4 | 24.9 |
| | Mean | 63.46 | **28.21** | 66.51 | **90.57** | 33.08 | **71.04** |

Table 7: Vision OOD example detection for the maximum softmax probability (MSP) baseline detector and the MSP detector after fine-tuning with Outlier Exposure (OE). All results are percentages and the result of 10 runs. Values are rounded so that $99.95\%$ rounds to $100\%$. More results are in Appendix E.

| $\mathcal{D}_{\text{in}}$ | $\mathcal{D}_{\text{out}}^{\text{test}}$ | FPR90 ↓ | | AUROC ↑ | | AUPR ↑ | |
|---|---|---|---|---|---|---|---|
| | | MSP | +OE | MSP | +OE | MSP | +OE |
| 20 Newsgroups | SNLI | 38.2 | 12.5 | 87.7 | 95.1 | 71.3 | 86.3 |
| | IMDB | 45.0 | 18.6 | 79.9 | 93.5 | 42.6 | 74.5 |
| | Multi30K | 54.5 | 3.2 | 78.3 | 97.3 | 45.8 | 93.7 |
| | WMT16 | 45.8 | 2.0 | 80.2 | 98.8 | 43.7 | 96.1 |
| | Yelp | 45.9 | 3.9 | 78.7 | 97.8 | 38.1 | 87.9 |
| | EWT-A | 36.1 | 1.2 | 86.2 | 99.2 | 58.2 | 97.3 |
| | EWT-E | 31.9 | 1.4 | 87.8 | 99.2 | 60.3 | 97.2 |
| | EWT-N | 41.7 | 1.8 | 83.1 | 98.7 | 46.2 | 95.7 |
| | EWT-R | 40.7 | 1.7 | 83.5 | 98.9 | 53.4 | 96.6 |
| | EWT-W | 44.5 | 2.4 | 81.1 | 98.5 | 39.0 | 93.8 |
| | Mean | 42.44 | **4.86** | 82.66 | **97.71** | 49.85 | **91.91** |
| TREC | SNLI | 18.2 | 4.2 | 94.0 | 98.1 | 81.9 | 91.6 |
| | IMDB | 49.6 | 0.6 | 78.0 | 99.4 | 44.2 | 97.8 |
| | Multi30K | 44.2 | 0.3 | 81.6 | 99.7 | 44.9 | 99.0 |
| | WMT16 | 50.7 | 0.2 | 78.2 | 99.8 | 42.2 | 99.4 |
| | Yelp | 50.9 | 0.4 | 75.1 | 99.7 | 37.7 | 96.1 |
| | EWT-A | 45.7 | 0.9 | 82.4 | 97.7 | 53.1 | 96.1 |
| | EWT-E | 36.8 | 0.4 | 85.7 | 99.5 | 60.8 | 99.1 |
| | EWT-N | 44.3 | 0.3 | 84.2 | 99.6 | 58.8 | 99.2 |
| | EWT-R | 46.1 | 0.4 | 82.5 | 99.5 | 51.1 | 98.8 |
| | EWT-W | 50.1 | 0.2 | 79.8 | 99.7 | 47.8 | 99.4 |
| | Mean | 43.46 | **0.78** | 82.14 | **99.28** | 52.23 | **97.64** |
| SST | SNLI | 57.3 | 33.4 | 75.7 | 86.8 | 36.2 | 52.0 |
| | IMDB | 83.1 | 32.6 | 54.3 | 85.9 | 19.0 | 51.5 |
| | Multi30K | 81.3 | 33.0 | 58.5 | 88.3 | 21.4 | 58.9 |
| | WMT16 | 76.0 | 17.1 | 60.2 | 92.9 | 21.4 | 68.8 |
| | Yelp | 82.0 | 11.3 | 54.2 | 92.7 | 18.8 | 60.0 |
| | EWT-A | 72.6 | 33.6 | 62.7 | 87.2 | 21.4 | 53.8 |
| | EWT-E | 68.1 | 26.5 | 68.5 | 90.4 | 27.0 | 63.7 |
| | EWT-N | 73.8 | 27.2 | 63.8 | 90.1 | 22.6 | 62.0 |
| | EWT-R | 79.6 | 41.4 | 58.1 | 85.6 | 20.3 | 54.7 |
| | EWT-W | 74.8 | 17.2 | 60.3 | 92.8 | 21.2 | 66.9 |
| | Mean | 74.86 | **25.31** | 61.61 | **90.07** | 22.93 | **64.37** |

Table 8: NLP OOD example detection for the maximum softmax probability (MSP) baseline detector and the MSP detector after fine-tuning with Outlier Exposure (OE). All results are percentages and the result of 10 runs. Values are rounded so that $99.95\%$ rounds to $100\%$.

**Anomalous Data.**    For each in-distribution dataset $\mathcal{D}_{\text{in}}$, we comprehensively evaluate OOD detectors on artificial and real anomalous distributions $\mathcal{D}_{\text{out}}^{\text{test}}$ following Hendrycks & Gimpel (2017). For each learned distribution $\mathcal{D}_{\text{in}}$, the number of test distributions that we compare against is approximately double that of most previous works.

*Gaussian* anomalies have each dimension i.i.d. sampled from an isotropic Gaussian distribution. *Rademacher* anomalies are images where each dimension is $-1$ or $1$ with equal probability, so each dimension is sampled from a symmetric Rademacher distribution. *Bernoulli* images have each pixel sampled from a Bernoulli distribution if the input range is $[0, 1]$. *Blobs* data consist in algorithmically generated amorphous shapes with definite edges. *Icons-50* is a dataset of icons and emojis (Hendrycks & Dietterich, 2019); icons from the "Number" class are removed. *Textures* is a dataset of describable textural images (Cimpoi et al., 2014). *Places365* consists in images for scene recognition rather than object recognition (Zhou et al., 2017). *LSUN* is another scene understanding dataset with fewer classes than Places365 (Yu et al., 2015). *ImageNet* anomalous examples are taken from the 800 ImageNet-1K classes disjoint from Tiny ImageNet's 200 classes, and when possible each image is cropped with bounding box information as in Tiny ImageNet. For the Places365 experiment, ImageNet is ImageNet-1K with all 1000 classes. With *CIFAR-10* as $\mathcal{D}_{\text{in}}$, we use also *CIFAR-100* as $\mathcal{D}_{\text{out}}^{\text{test}}$ and vice versa; recall that the CIFAR-10 and CIFAR-100 classes do not overlap. *Chars74K* is

a dataset of photographed characters in various styles; digits and letters such as "O" and "l" were removed since they can look like numbers. *Places69* has images from 69 scene categories not found in the Places365 dataset.

*SNLI* is a dataset of predicates and hypotheses for natural language inference. We use the hypotheses for $\mathcal{D}_{\text{out}}^{\text{OE}}$. *IMDB* is a sentiment classification dataset of movie reviews, with similar statistics to those of SST. *Multi30K* is a dataset of English-German image descriptions, of which we use the English descriptions. *WMT16* is the English portion of the test set from WMT16. *Yelp* is a dataset of restaurant reviews. *English Web Treebank (EWT)* consists of five individual datasets: Answers (A), Email (E), Newsgroups (N), Reviews (R), and Weblog (W). Each contains examples from the indicated domain.

**Validation Data.** For each experiment, we create a set of validation distributions $\mathcal{D}_{\text{out}}^{\text{val}}$. The first anomalies are *uniform noise* anomalies where each pixel is sampled from $\mathcal{U}[0, 1]$ or $\mathcal{U}[-1, 1]$ depending on the input space of the classifier. The remaining $\mathcal{D}_{\text{out}}^{\text{val}}$ validation sources are generated by corrupting in-distribution data, so that the data becomes out-of-distribution. One such source of anomalies is created by taking the pixelwise *arithmetic mean* of a random pair of in-distribution images. Other anomalies are created by taking the *geometric mean* of a random pair of in-distribution images. *Jigsaw* anomalies are created by taking an in-distribution example, partitioning the image into 16 equally sized patches, and permuting those patches. *Speckle Noised* anomalies are created by applying speckle noise to in-distribution images. *RGB Ghosted* anomalies involves shifting and reordering the color channels of in-distribution images. *Inverted* images are anomalies which have some or all of their color channels inverted.

# B ARCHITECTURES AND TRAINING DETAILS

For CIFAR-10, CIFAR-100, and Tiny ImageNet classification experiments, we use a 40-2 Wide Residual Network (Zagoruyko & Komodakis, 2016). The network trains for 100 epochs with a dropout rate of 0.3. The initial learning rate of 0.1 decays following a cosine learning rate schedule (Loshchilov & Hutter, 2017). During fine-tuning of the entire network, we again use a cosine learning rate schedule but with an initial learning rate of 0.001. We use standard flipping and data cropping augmentation, Nesterov momentum, and $\ell_2$ weight decay with a coefficient of $5 \times 10^{-4}$. SVHN architectures are 16-4 Wide ResNets trained for 20 epochs with an initial learning rate of 0.01 and no data augmentation. For Places365, we use a ResNet-18 pre-trained on Places365. In this Places365 experiment, we tune with Outlier Exposure for 5 epochs, use 512 outlier samples per iteration, and start with a learning rate of 0.0001. Outlier Exposure fine-tuning occurs with each epoch being the length of in-distribution dataset epoch, so that Outlier Exposure completes quickly and does involve reading the entire $\mathcal{D}_{\text{out}}^{\text{OE}}$ dataset.

# C TRAINING FROM SCRATCH WITH OUTLIER EXPOSURE USUALLY IMPROVES DETECTION PERFORMANCE

Elsewhere we show results for pre-trained networks that are fine-tuned with OE. However, a network trained from scratch which simultaneously trains with OE tends to give superior results. For example, a CIFAR-10 Wide ResNet trained normally obtains a classification error rate of 5.16% and an FPR95 of 34.94%. Fine-tuned, this network has an error rate of 5.27% and an FPR95 of 9.50%. Yet if we instead train the network from scratch and expose it to outliers as it trains, then the error rate is 4.26% and the FPR95 is 6.15%. This architecture corresponds to a 9.50% RMS calibration error with OE fine-tuning, but by training with OE from scratch the RMS calibration error is 6.15%. Compared to fine-tuning, training a network in tandem with OE tends to produce a network with a better error rate, calibration, and OOD detection performance. The reason why we use OE for fine-tuning is because training from scratch requires more time and sometimes more GPU memory than fine-tuning.

# D OE WORKS ON OTHER VISION ARCHITECTURES

Outlier Exposure also improves vision OOD detection performance for more than just Wide ResNets. Table 9 shows that Outlier Exposure also improves vision OOD detection performance for "All Convolutional Networks" (Salimans & Kingma, 2016).

| $\mathcal{D}_{\text{in}}$ | FPR95 ↓ | | AUROC ↑ | | AUPR ↑ | |
|---|---|---|---|---|---|---|
| | MSP | +OE | MSP | +OE | MSP | +OE |
| SVHN | 6.84 | 0.08 | 98.1 | 100.0 | 90.9 | 99.8 |
| CIFAR-10 | 28.4 | 14.0 | 90.1 | 96.7 | 58.9 | 87.3 |
| CIFAR-100 | 57.5 | 43.3 | 76.7 | 85.3 | 33.9 | 51.3 |
| Tiny ImageNet | 75.5 | 25.0 | 55.4 | 82.9 | 25.6 | 75.3 |

Table 9: Results using an All Convolutional Network architectures. Results are percentages and an average of 10 runs.

## E  OUTLIER EXPOSURE WITH $H(\mathcal{U}; p)$ SCORES DOES BETTER THAN WITH MSP SCORES

While $-\max_c f_c(x)$ tends to be a discriminative OOD score for example $x$, models with OE can do better by using $-H(\mathcal{U}; f(x))$ instead. This alternative accounts for classes with small probability mass rather than just the class with most mass. Additionally, the model with OE is trained to give anomalous examples a uniform posterior not just a lower MSP. This simple change roundly aids performance as shown in Table 10. This general performance improvement is most pronounced on datasets with many classes. For instance, when $\mathcal{D}_{\text{out}}^{\text{test}} = $ Tiny ImageNet and $\mathcal{D}_{\text{out}}^{\text{test}} = $ Gaussian, swapping the MSP score with the $H(\mathcal{U}; f(x))$ score increases the AUROC 76.5% to 97.1%.

| $\mathcal{D}_{\text{in}}$ | FPR95 ↓ | | AUROC ↑ | | AUPR ↑ | |
|---|---|---|---|---|---|---|
| | MSP | $H(\mathcal{U}; p)$ | MSP | $H(\mathcal{U}; p)$ | MSP | $H(\mathcal{U}; p)$ |
| CIFAR-10 | 9.50 | 9.04 | 97.81 | 97.92 | 90.48 | 90.85 |
| CIFAR-100 | 38.50 | 33.31 | 87.89 | 88.46 | 58.15 | 58.30 |
| Tiny ImageNet | 13.99 | 7.45 | 92.18 | 95.45 | 79.26 | 85.71 |
| Places365 | 28.21 | 19.58 | 90.57 | 92.53 | 71.04 | 74.39 |

Table 10: Comparison between the maximum softmax probability (MSP) and $H(\mathcal{U}; p)$ OOD scoring methods on a network fine-tuned with OE. Results are percentages and an average of 10 runs. For example, CIFAR-10 results are averaged over "Gaussian," "Rademacher," ..., or "CIFAR-100" measurements.

## F  EXPANDED LANGUAGE MODELING RESULTS

Detailed OOD detection results with language modeling datasets are shown in Table 11.

| $\mathcal{D}_{\text{in}}$ | $\mathcal{D}_{\text{out}}^{\text{test}}$ | FPR90 ↓ | | AUROC ↑ | | AUPR ↑ | |
|---|---|---|---|---|---|---|---|
| | | BPC | +OE | BPC | +OE | BPC | +OE |
| PTB Char | Answers | 96.9 | 49.93 | 82.1 | 89.6 | 81.0 | 89.3 |
| | Email | 99.5 | 90.64 | 80.6 | 88.6 | 79.4 | 89.1 |
| | Newsgroup | 99.8 | 99.39 | 75.2 | 85.0 | 73.3 | 85.5 |
| | Reviews | 99.0 | 74.64 | 80.8 | 89.0 | 79.2 | 89.6 |
| | Weblog | 100.0 | 100.0 | 68.9 | 79.2 | 67.3 | 80.1 |
| Mean | | 99.0 | **89.4** | 77.5 | **86.3** | 76.0 | **86.7** |
| PTB Word | Answers | 41.4 | 3.65 | 81.4 | 98.0 | 40.5 | 94.7 |
| | Email | 64.9 | 0.17 | 78.1 | 99.6 | 44.5 | 98.9 |
| | Newsgroup | 54.9 | 0.17 | 77.8 | 99.5 | 39.8 | 98.3 |
| | Reviews | 30.5 | 0.85 | 88.0 | 98.9 | 53.6 | 96.8 |
| | Weblog | 50.8 | 0.08 | 80.7 | 99.9 | 41.5 | 99.7 |
| Mean | | 48.5 | **0.98** | 81.2 | **99.2** | 44.0 | **97.8** |

Table 11: OOD detection results on Penn Treebank examples and English Web Treebank outliers. All results are percentages.

The $\mathcal{D}_{\text{out}}^{\text{test}}$ datasets come from the English Web Treebank (Bies et al., 2012), which contains text from five different domains: Yahoo! Answers, emails, newsgroups, product reviews, and weblogs. Other

NLP $\mathcal{D}_{\text{out}}^{\text{test}}$ datasets we consider do not satisfy the language modeling assumption of continuity in the examples, so we do not evaluate on them.

## G  CONFIDENCE CALIBRATION

Models integrated into a decision making process should indicate when they are trustworthy, and such models should not have inordinate confidence in their predictions. In an effort to combat a false sense of certainty from overconfident models, we aim to calibrate model confidence. A model is calibrated if its predicted probabilities match empirical frequencies. Thus if a calibrated model predicts an event with 30% probability, then 30% of the time the event transpires. Prior research (Guo et al., 2017; Nguyen & O'Connor, 2015; Kuleshov & Liang, 2015) considers calibrating systems where test-time queries are samples from $\mathcal{D}_{\text{in}}$, but systems also encounter samples from $\mathcal{D}_{\text{out}}^{\text{test}}$ and should also ascribe low confidence to these samples. Hence, we use OE to control the confidence on these samples.

### G.1  METRICS

In order to evaluate a multiclass classifier's calibration, we present three metrics. First we establish context. For input example $X \in \mathcal{X}$, let $Y \in \mathcal{Y} = \{1, 2, \ldots, k\}$ be the ground truth class. Let $\widehat{Y}$ be the model's class prediction, and let $C$ be the corresponding model confidence or prediction probability. Denote the set of prediction-label pairs made by the model with $S = \{(\widehat{y}_1, c_1), (\widehat{y}_2, c_2), \ldots, (\widehat{y}_n, c_n)\}$.

**RMS and MAD Calibration Error.**  The Root Mean Square Calibration Error measures the square root of the expected squared difference between confidence and accuracy at a confidence level. It has the formula $\sqrt{\mathbb{E}_C[(\mathbb{P}(Y = \widehat{Y}|C = c) - c)^2]}$. A similar formulation which less severely penalizes large confidence-accuracy deviations is the Mean Absolute Value Calibration error, written $\mathbb{E}_C[|\mathbb{P}(Y = \widehat{Y}|C = c) - c|]$. The MAD Calibration Error is a lower bound of the RMS Calibration Error. To empirically estimate these miscalibration measures, we partition the $n$ samples of $S$ into $b$ bins $\{B_1, B_2, \ldots, B_b\}$ with approximately 100 samples in each bin. Unlike Guo et al. (2017), bins are not equally spaced since the distribution of confidence values is not uniform but dynamic. Concretely, the RMS Calibration Error is estimated with the numerically stable formula

$$\sqrt{\sum_{i=1}^{b} \frac{|B_i|}{n} \left( \frac{1}{|B_i|} \sum_{k \in B_i} \mathbb{1}(y_k = \widehat{y}_k) - \frac{1}{|B_i|} \sum_{k \in B_i} c_k \right)^2}.$$

Along similar lines, the MAD Calibration Error—which is an improper scoring rule due to its use of absolute differences rather than squared differences—is estimated with

$$\sum_{i=1}^{b} \frac{|B_i|}{n} \left| \frac{1}{|B_i|} \sum_{k \in B_i} \mathbb{1}(y_k = \widehat{y}_k) - \frac{1}{|B_i|} \sum_{k \in B_i} c_k \right|.$$

**Soft F1 Score.**  If a classifier makes only a few mistakes, then most examples should have high confidence. But if the classifier gives all predictions high confidence, including its mistakes, then the previous metrics will indicate that the model is calibrated on the vast majority of instances, despite having systematic miscalibration. The Soft F1 score (Pastor-Pellicer et al., 2013; Hendrycks & Gimpel, 2017) is suited for measuring the calibration of a system where there is an acute imbalance between mistaken and correct decisions. Since we treat mistakes a positive examples, we can write the model's confidence that the examples are anomalous with $c_a = (1 - c_1, 1 - c_2, \ldots, 1 - c_n)$. To indicate that an example is positive (mistaken), we use the vector $m \in \{0, 1\}^n$ such that $m_i = \mathbb{1}(y_i \neq \widehat{y}_i)$ for $1 \leq i \leq n$. Then the Soft F1 score is

$$\frac{c_a^{\mathsf{T}} m}{\mathbf{1}^{\mathsf{T}}(c_a + m)/2}.$$

| $\mathcal{D}_{\text{in}}$ | RMS Calib. Error ↓ | | MAD Calib. Error ↓ | | Soft F1 Score ↑ | |
|---|---|---|---|---|---|---|
| | Temperature | +OE | Temperature | +OE | Temperature | +OE |
| SVHN | 22.3 | 4.8 | 10.9 | 2.4 | 52.1 | 87.9 |
| CIFAR-10 | 19.4 | 9.3 | 12.6 | 5.6 | 39.9 | 69.7 |
| CIFAR-100 | 14.7 | 8.2 | 11.3 | 6.5 | 52.8 | 65.8 |
| Tiny ImageNet | 12.0 | 6.9 | 9.0 | 4.8 | 62.9 | 72.2 |

Table 12: Calibration results for the temperature tuned baseline and temperature tuning + OE.

### G.2 SETUP AND RESULTS

There are many ways to estimate a classifier's confidence. One way is to bind a logistic regression branch onto the network, so that confidence values are in $[0, 1]$. Other confidence estimates use the model's logits $l \in \mathbb{R}^k$, such as the estimate $\sigma(\max_i l_i) \in [0, 1]$, where $\sigma$ is the logistic sigmoid. Another common confidence estimate is $\max_i \left[ \exp(l_i) / \sum_{j=1}^{k} \exp(l_j) \right]$. A modification of this estimate is our baseline.

**Softmax Temperature Tuning.** Guo et al. (2017) show that good calibration can be obtained by including a tuned temperature parameter into the softmax: $\widehat{p}(y = i \mid x) = \exp(l_i/T) / \sum_{j=1}^{k} \exp(l_j/T)$. We tune $T$ to maximize log likelihood on a validation set after the network has been trained on the training set.

**Results.** In this calibration experiment, the baseline is confidence estimation with softmax temperature tuning. Therefore, we train SVHN, CIFAR-10, CIFAR-100, and Tiny ImageNet classifiers with 5000, 5000, 5000, and 10000 training examples held out, respectively. A copy of this classifier is fine-tuned with Outlier Exposure. Then we determine the optimal temperatures of the original and OE-fine-tuned classifiers on the held-out examples. To measure calibration, we take equally many examples from a given in-distribution dataset $\mathcal{D}_{\text{in}}^{\text{test}}$ and OOD dataset $\mathcal{D}_{\text{out}}^{\text{test}}$. Out-of-distribution points are understood to be incorrectly classified since their label is not in the model's output space, so calibrated models should assign these out-of-distribution points low confidence. Results are in Table 12. Outlier Exposure noticeably improves model calibration.

### G.3 POSTERIOR RESCALING

While temperature tuning improves calibration, the confidence estimate $\widehat{p}(y = i \mid x)$ cannot be less than $1/k$, $k$ the number of classes. For an out-of-distribution example like Gaussian Noise, a good model should have no confidence in its prediction over $k$ classes. One possibility is to add a reject option, or a $(k+1)$st class, which we cover in Section 5. A simpler option we found is to perform an affine transformation of $\widehat{p}(y = i \mid x) \in [1/k, 1]$ with the formula $(\widehat{p}(y = i \mid x) - 1/k)/(1 - 1/k) \in [0, 1]$. This simple transformation makes it possible for a network to express no confidence on an out-of-distribution input and improves calibration performance. As Table 13 shows, this simple 0-1 posterior rescaling technique consistently improves calibration, and the model fine-tuned with OE using temperature tuning and posterior rescaling achieved large calibration improvements.

| $\mathcal{D}_{\text{in}}$ | RMS Calib. Error ↓ | | | MAD Calib. Error ↓ | | | Soft F1 Score ↑ | | |
|---|---|---|---|---|---|---|---|---|---|
| | Temp | +Rescale | +OE | Temp | +Rescale | +OE | Temp | +Rescale | +OE |
| SVHN | 22.3 | 20.8 | 3.0 | 10.9 | 10.1 | 1.0 | 52.1 | 56.1 | 92.7 |
| CIFAR-10 | 19.4 | 17.8 | 6.7 | 12.6 | 11.7 | 4.1 | 39.9 | 42.8 | 73.9 |
| CIFAR-100 | 14.7 | 14.4 | 8.1 | 11.3 | 11.1 | 6.4 | 52.8 | 53.1 | 66.1 |
| Tiny ImageNet | 12.0 | 11.9 | 6.9 | 9.0 | 8.8 | 4.8 | 62.9 | 63.1 | 72.3 |

Table 13: Calibration results for the softmax temperature tuning baseline, the same baseline after adding Posterior Rescaling, and temperature tuning + Posterior Rescaling + OE.

# H    ADDITIONAL ROC AND PR CURVES

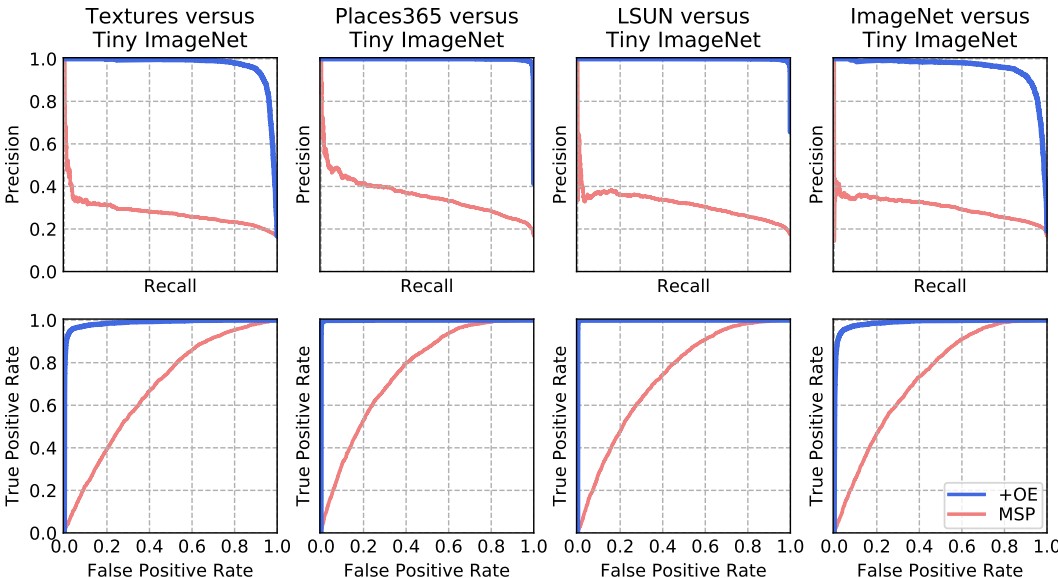

Figure 4: ROC curves with Tiny ImageNet as $\mathcal{D}_{in}$ and Textures, Places365, LSUN, and ImageNet as $\mathcal{D}_{out}^{test}$. Figures show the curves corresponding to the maximum softmax probability (MSP) baseline detector and the MSP detector with Outlier Exposure (OE).

In Figure 4, we show additional PR and ROC Curves using the Tiny ImageNet dataset and various anomalous distributions.

