# OpenReview forum: "Deep Anomaly Detection with Outlier Exposure"
_ICLR.cc/2019/Conference_

### Official Review · AnonReviewer2 · 2018-10-28
**A comprehensive study of an intuitive idea for anomaly detection. Can benefit from a restructuring of the writing.**

**Rating:** 8
**Confidence:** 4

**Review:**

This paper proposes fine-tuning an out-of-distribution detector using an Outlier Exposure (OE) dataset. The novelty is in proposing a model-specific rather than dataset-specific fine-tuning. Their modifications are referred to as Outlier Exposure. OE includes the choice of an OE dataset for fine-tuning and a regularization term evaluated on the OE dataset. It is a comprehensive study that explores multiple datasets and improves dataset-specific baselines.

Suggestions and clarification requests:
- The structure of the writing does not clearly present the novel aspects of the paper as opposed to the previous works. I suggest moving the details of model-specific OE regularization terms to section 3 and review the details of the baseline models. Then present the other set of novelties in proposing OE datasets in a new section before presenting the results. Clearly presenting two sets of novelties in this work and then the results. If constrained in space, I suggest squeezing the discussion, conclusion, and 4.1.
- In the related work section Radford et al., 2016 is references when mentioning GAN. Why not the original reference for GAN?
- Maybe define BPP, BPC, and BPW in the paragraphs on PixelCNN++ and language modeling or add a reference.
- Numbers in Table 3 column MSP should match the numbers in Table 1, right? Or am I missing something?

---

> ### Author Response · Authors · 2018-11-21
> **Uploaded Draft Adding the Suggestions**
>
> Thank you for your careful analysis of our paper.
>
> We have uploaded a new draft incorporating your suggestions.
>
> To improve clarity, we have added two paragraphs to the preface of Section 4 summarizing our experiments and novel discoveries. We found it difficult to import several specific details from individual experiments to Section 3, so we opted to instead improve the clarity of several experimental sections as they appear, and to improve the clarity of the discussion section. We also restructured the calibration section.
>
> Regarding your second and third points, we added the reference for the original GAN paper, and we added definitions for BPP, BPC, and BPW to Section 4.4. Thank you for these suggestions.
>
> The baseline numbers in Table 3 differ from those in Table 1 because in Table 3 we use the training regime from the publicly available implementation of DeVries et al. to create an accurate comparison. The difference is that they use a different learning schedule than the models from Table 1.

---

### Official Review · AnonReviewer1 · 2018-11-05
**An outlier detection method that assumes access to the outlier distribution?**

**Rating:** 6
**Confidence:** 4

**Review:**

This paper describes how a deep neural network can be fine-tuned to perform outlier detection in addition to its primary objective. For classification, the fine-tuning objective encourages out-of-distribution samples to have a uniform distribution over all class labels. For density estimation, the objective encourages out-of-distribution samples to be ranked as less probability than in-distribution samples. On a variety of image and text datasets, this additional fine-tuning step results in a network that does much better at outlier detection than a naive baseline, sometimes approaching perfect AUROC.

The biggest weakness in this paper is the assumption that we have access to out-of-distribution data, and that we will encounter data from that same distribution in the future. For the typical anomaly detection setting, we expect that anomalies could look like almost anything. For example, in network intrusion detection (a common application of anomaly detection), future attacks are likely to have different characteristics than past attacks, but will still look unusual in some way. The challenge is to define "normal" behavior in a way that captures the full range of normal while excluding "unusual" examples. This topic has been studied for decades.

Thus, I would not classify this paper as an anomaly detection paper. Instead, it's defining a new task and evaluating performance on that task. The empirical results demonstrate that the optimization succeeds in optimizing the objective it was given. What's missing is the justification for this problem setting -- when is it the case that we need to detect outliers *and* have access to the distribution over outliers?

--------

UPDATE AFTER RESPONSE PERIOD:

My initial read of this paper was incorrect -- the authors do indeed separate the outlier distribution used to train the detector from the outlier distribution used for evaluation. Much of these details are in Appendix A; I suggest that the authors move some of this earlier or more heavily reference Appendix A when describing the methods and introducing the results. I am not well-read in the other work in this area, but this looks like a nice advance.

Based on my read of the related work section (again, having not studied the other papers), it looks like this work fills a slightly different niche from some previous work. In particular, OE is unlikely to be adversarially robust. So this might be a poor choice for finding anomalies that represent malicious behavior (e.g., network intrusion detection, adversarial examples, etc.), but good for finding natural examples from a different distribution (e.g., data entry errors).

My main remaining reservation is that this work is still at the stage of empirical observation -- I hope that future work (by these authors or others) can investigate the assumptions necessary for this method to work, and even characterize how well we should expect it to work. Without a framework for understanding generalization in this context, we may see a proliferation of heuristics that succeed on benchmarks without developing the underlying principles.

---

> ### Author Response · Authors · 2018-11-21
> **Clarifying the Problem Setup**
>
> Thank you for your thoughtful feedback and willingness to question the premises behind submitted works.
>
> We believe there may be a misunderstanding of our experimental setup. In the setup you describe, out-of-distribution data is available during training, and data from that same distribution is encountered at test time. We agree that such a setup has issues, and we intentionally avoided that setup. We do not assume access to the test distribution, but this confusion is understandable as many recent OOD papers assume this. In particular, we took great care to keep datasets disjoint in our experiments, and the only out-of-distribution dataset examples we use at training time come from the realistic, diverse Outlier Exposure datasets described in Section 4.2.2. We ensured that these OE datasets were disjoint with the out-of-distribution data evaluated at test time. For instance, in the NLP experiments, we used WikiText-2 as the OE dataset, and none of the NLP OOD datasets evaluated on at test time were collected from Wikipedia.
>
> One of our contributions is that training on the OE datasets which we identified leads to generalization to novel forms of anomalies. Concretely, with SVHN as the in-distribution, we found that OE improved OOD detection on the Icons-50 dataset of emojis, even though the OE dataset consisted in natural images and did not contain any emojis. Thus, training with OE does help with generalization to new anomalies, and it does not simply teach the detector a particular, narrow distribution of outliers.

---

> ### Author Response · Authors · 2018-11-27
> **On Not Accessing Test Data**
>
> Reviewer 1, we have added more emphasis that the Outlier Exposure data and the test sets are disjoint in the revised draft.

---

> > ### Comment · AnonReviewer1 · 2018-11-28
> > **Additional clarification**
> >
> > Thanks for the clarification. Yes, it makes a big difference that the "training" outliers are from different datasets than the "test" outliers -- I'm happy I was mistaken in my previous understanding.
> >
> > I'll study the paper some more, but after quickly rereading some key sections, I don't understand exactly what combinations of D_in, D_out^OE, and D_out^test were used, e.g., in Table 1. From the row labels, I can figure out what D_in is. From Section 4.2.2., it sounds like you used 80 Million Tiny Images as D_out^OE for SVHN, CIFAR10, and CIFAR-100. Was ImageNet-22K used as D_out^OE for Tiny ImageNet? The text is ambiguous. And then, what was used for D_out^test?
> >
> > In general, the effectiveness of these techniques will rely heavily on the nature of the datasets used. With some combinations, we should expect OE to reduce the accuracy of anomaly detection, much like the "negative transfer" phenomenon in transfer learning. I didn't see much discussion of this point, but perhaps I missed it.

---

> > > ### Author Response · Authors · 2018-11-28
> > > **Follow-up**
> > >
> > > Thank you for your reply and good questions. Due to space limitations, in Appendix A we list the results for each D_out^test distribution, and we give the full descriptions of the D_out^test distributions. The test D_out^test distributions consist in Gaussian Noise, Rademacher Noise, Bernoulli Noise, Blobs, Icons-50 (emojis), Textures, Places365, LSUN, ImageNet (the 800 ImageNet-1K classes not in Tiny ImageNet and not in D_out^OE), CIFAR-10/100, and Chars74K anomalies. For NLP we use SNLI, IMDB, Multi30K, WMT16, Yelp, and various subsets of the English Web Treebank. We therefore test our models with approximately double the number of D_out^test image distributions compared to prior work; we also test in NLP, unlike nearly all other recent work in OOD detection.
> > >
> > > Your read is correct that 80 Million Tiny Images are used for SVHN, CIFAR-10, CIFAR-100; these images are too low-resolution (32x32x3) for Tiny ImageNet, so for that we use ImageNet-22K (minus ImageNet-1K). For NLP, we use WikiText-2, but in the discussion we note using the Project Gutenberg corpus also works, so the dataset choice has flexibility even in NLP. Thanks to your comment, we will add a link to Appendix A in the caption of Table 1 for the full results and make the interactions between D_in, D_out^OE, D_out^test clearer.
> > >
> > > As for accuracy, the fixed coefficient of lambda = 0.5 for the vision experiments leads to slight degradation when tuning with OE, like other approaches. For example, a vanilla CIFAR-10 Wide ResNet has 5.16% classification error, while with OE tuning it has 5.27% error. This degradation can be further reduced by training from scratch (Appendix E). We will look into ``negative transfer.'' Thank you.

---

### Official Review · AnonReviewer3 · 2018-11-10
**Research topic is interesting, but the paper needs improvement.**

**Rating:** 6
**Confidence:** 5

**Review:**

I have read authors' reply.  In response to authors' comprehensive reply and feedback. I upgrade my score to 6. As authors mentioned, the extension to density estimators is an original novelty of this paper, but I still have some concern that OE loss for classification is basically the same as [2]. I think it is better to clarify this in the draft.

Summary===

This paper proposes a new fine-tuning method for improving the performance of existing anomaly detectors. The main idea is additionally optimizing the “Outlier Exposure (OE)” loss on outlier dataset. Specifically, for softmax classifier, the authors set the OE loss to the KL divergence loss between posterior distribution and uniform distribution. For density estimator, they set the OE loss to a margin ranking loss. The proposed method improves the detection performance of baseline methods on various vision and NLP datasets. While the research topic of this paper is interesting, I recommend rejections because I have concerns about novelty and the experimental results.

Detailed comments ===

1. OE loss for softmax classifier

For softmax classifier, the OE loss forces the posterior distribution to become uniform distribution on outlier dataset. I think this loss function is very similar to a confidence loss (equation 2) proposed in [2]: Lee et al., 2017 [2] also proposed the loss function minimizing the KL divergence between posterior distribution and uniform distribution on out-of-distribution, and evaluated the effects of it on "unseen" out-of-distribution (see Table 1 of [2]). Could the authors clarify the difference with the confidence loss in [2], and compare the performance with it? Without that, I feel that the novelty of this paper is not significant.

2. More comparison with baselines

The authors said that they didn’t compare the performance with simple inference methods like ODIN [3] since ODIN tunes the hyper-parameters using data from (tested) out-of-distribution. However, I think that the authors can compare the performance with ODIN by tuning the hyper-parameters of it on outlier dataset which is used for training OE loss. Could the authors provide more experimental results by comparing the performance with ODIN?

3. Related work

I would appreciate if the authors can survey and compare more baselines such as [4] and [5].

[1] Dan Hendrycks and Kevin Gimpel. A baseline for detecting misclassified and out-of-distribution examples in neural networks. International Conference on Learning Representations, 2017.
[2] Kimin Lee, Honglak Lee, Kibok Lee, and Jinwoo Shin. Training confidence-calibrated classifiers for detecting out-of-distribution samples. International Conference on Learning Representations, 2018.
[3] Shiyu Liang, Yixuan Li, and R. Srikant. Enhancing the reliability of out-of-distribution image detection in neural networks. International Conference on Learning Representations, 2018.
[4] Kimin Lee, Kibok Lee, Honglak Lee, and Jinwoo Shin. A Simple Unified Framework for Detecting Out-of-Distribution Samples and Adversarial Attacks. In NIPS, 2018.
[5] Apoorv Vyas, Nataraj Jammalamadaka, Xia Zhu, Dipankar Das, Bharat Kaul, and Theodore L. Willke. Out-of-Distribution Detection Using an Ensemble of Self Supervised Leave-out Classifiers, In ECCV, 2018.

---

> ### Author Response · Authors · 2018-11-20
> **Comparison Given and ODIN Results Added**
>
> Thank you for your detailed feedback.
>
> 1.
> Lee et al. [2] propose training against GAN-generated out-of-distribution data, and they use a confidence loss for anomaly detection with multiclass classification as the original task. By contrast, we consider a broader range of original tasks, including density estimation and natural language settings, and we show how to incorporate Outlier Exposure for each scenario.
>
> Another crucial difference between our work and [2] is that we demonstrate that realistic, diverse data is significantly more effective than GAN-generated examples, and is scalable to complex, high-resolution data that everyday GANs have difficulty generating. Likewise, GANs are currently not capable of generating high-quality text. Finally, Lee et al. [2] state in Appendix B, “For each out-of-distribution dataset, we randomly select 1,000 images for tuning the penalty parameter β, mini-batch size and learning rate.” Thus some of their hyperparameters are tuned on OOD test data, which is not the case in our work. Hence, our work is in a different setting from Lee et al. [2]. In our paper we show how to use real data to _consistently_ improve detection in a host of settings. In essence, our some of our multiclass experiments are built on the seminal work of Lee et al. [2] by using real and diverse data.
>
> Our primary contribution is that real data from a diverse source can be used to train anomaly detectors which generalize to anomalies from new and different distributions, so there is no need to use GANs or assume access to the test distributions. We demonstrate this in a variety of settings, showing that this technique is general and consistently boosts performance.
>
> Secondary sources of novelty in our paper include the margin loss for OOD detection with density estimators, the cross entropy OOD score instead of MSP (Appendix G), posterior rescaling for confidence calibration in the presence of OOD data (Appendix C), and our observation that a cutting-edge CIFAR-10 density model unexpectedly assigns higher density to SVHN images than to CIFAR-10 images. The latter contribution forms the basis for a concurrent submission by different authors, which can be found here: https://openreview.net/forum?id=H1xwNhCcYm Since that work is concurrent, it does not detract from our paper’s novelty. We should note that we not only reveal that density estimates are unreasonable on out-of-distribution points, but we also ameliorate it with Outlier Exposure.
>
> 2.
> We have added a section comparing to ODIN [3] (Appendix I). We will incorporate the results into the main paper if you think we should.
>
> 3.
> Thank you for pointing out these related works. The works of [4] and [5] are ECCV 2018 and NIPS 2018 papers, both of which are for conferences occurring after the submission deadline of this paper. We have a working implementation of [4] and will incorporate it into the paper it once we are sure that it is a faithful reproduction. We think that our comparisons on multiclass OOD detection (including the baseline [1], Lee et al. [2], DeVries et al., Liang et al. [3]), density estimation OOD detection, and confidence calibration on vision and NLP datasets are sufficient to demonstrate our method.
>
> Edit: Thank you very much for taking the time to read this response and update your score.

---

### Author Response · Authors · 2018-10-01
**Parallel Work**

In Section 4.3 we observe that a cutting-edge CIFAR-10 density model unexpectedly assigns higher density to SVHN images than to CIFAR-10 images.
As it happens, a concurrent submission is based on this observation. Their work can be found here: https://openreview.net/forum?id=H1xwNhCcYm

---

### Public Comment · ~Andrey_Malinin1 · 2018-10-10
**Related Work**

Hello! :) Interesting work. You may find our work on predictive uncertainty estimation to be relevant relevant.

https://arxiv.org/pdf/1802.10501.pdf

---

> ### Author Response · Authors · 2018-10-14
> **Reply**
>
> Thank you for bringing your NIPS 2018 paper to our attention. We think decoupling uncertainty into "data" and "OOD" uncertainty is an interesting avenue, and we will cite your work accordingly.

---

### Public Comment · (anonymous) · 2018-11-13
**Related work**

Hi,
we have a complementary out-of-distribution detection paper currently under review:
https://openreview.net/forum?id=H1x1noAqKX
We detect OOD samples on a pixel level. We also find that using outliers during training is effective for detecting OOD samples.

---

> ### Author Response · Authors · 2018-11-28
> **Interesting Task**
>
> This is an interesting segmentation task, and we will be sure to try Outlier Exposure on this task in the future. We intend to include a citation to your work after submission deanonymization.

---

> > ### Public Comment · (anonymous) · 2018-12-06
> > **Related work**
> >
> > Thanks! Of course, we will be happy to cite your work on the first occasion.

---

### Author Response · Authors · 2019-01-28
**Paper and Code Now Available**

We have put up a de-anonymized version of the paper. Unlike the draft from the reviewing cycle, this draft shows OE can also work on large-scale images (Places365). Code for most of the experiments, including the NLP experiments, has been made available: https://github.com/hendrycks/outlier-exposure

---

### Meta-Review · Area_Chair1 · 2018-12-12
**Limited novelty, but interesting results**

**Confidence:** 4
**Recommendation:** Accept (Poster)

**Metareview:**

The paper proposes a new fine-tuning method for improving the performance of existing anomaly detectors.

The reviewers and AC note the limitation of novelty beyond existing literature.

This is quite a borader line paper, but AC decided to recommend acceptance as comprehensive experimental results (still based on empirical observation though) are interesting.